# Hourly precipitation fields at 1 km resolution over Belgium from 1940 to 2016 based on the analog technique

Elke Debrie<sup>1</sup>, Jonathan Demaeyer<sup>1</sup>, and Stéphane Vannitsem<sup>1</sup>

<sup>1</sup>Meteorological and Climatological Information Service, Royal Meteorological Institute of Belgium, Brussels, Belgium

**Correspondence:** Stéphane Vannitsem (stephane.vannitsem@meteo.be)

**Abstract.** High-resolution gridded precipitation data is scarce, especially at time intervals shorter than daily. However hydrological applications for example benefit from a finer temporal resolution of rainfall information. In this context, we introduce an hourly precipitation dataset for Belgium, featuring a resolution of 1 km. An hourly high-resolution gridded precipitation product over Belgium can provide valuable insights into the dynamics of both short-term and long-term rainfall events, which can be used for wide-ranging applications such as flash flood forecasting and warning systems, studying precipitation extremes and trends, validating weather and climate models or detecting changes in precipitation patterns due to climate change. Similar products such as EURADCLIM (Europe) (Overeem et al., 2023) and RADKLIM (Germany) (Winterrath, 2018), both radar-based gauge-adjusted datasets, have already been created and published. Both datasets are high spatial resolution dataset (2 km and 1 km, respectively).

A high resolution precipitation grid of hourly precipitation data for Belgium covering the period from 1940 to 2016 using the analog technique, is created. The analogs are sampled from the period 2017-2022 for which high resolution radar data precipitation fields are available. The initial step involves identifying the criteria, i.e. atmospheric parameters such as atmospheric pressure, temperature and humidity, that can be used to determine analogous days. These atmospheric parameters are obtained from the ERA5 observational data provided by the European Centre for Medium-Range Weather Forecasts (ECMWF). In a second step, hourly precipitation data for suitable analog days are extracted from our radar database, and then used to create the high resolution grid of hourly precipitation for Belgium from 1940 to 2016. Data from rain gauges on the Belgian terrain were used for validation of the candidate precipitation analogs.

The dataset for this project lists the top 25 analog days for 1940-2016 based on similarities in weather patterns. The analogs are ranked based on how closely they match to their target day.

The database is relying on the *Zarr* archiving format and is composed of two archives. A first archive contains all target days together with the 25 best analogs. The second one provides a precipitation field for each hour of every day in the past. The Zarr format of the database allows slicing through the database. For example, it allows one to easily delimit a specific area

of interest and a specific time frame for which the high resolution gridded median hourly precipitation fields are needed. The *median field dataset* is available on Zenodo (https://doi.org/10.5281/zenodo.14965710) (Debrie et al., 2025).

## Copyright statement. TEXT

# 1 Introduction

25

Recently, increased attention has been given to sub-daily precipitation observations due to intense rainfall events such as during the recent flood of July 2021 which affected Belgium, Germany and Luxembourg (Tradowsky et al., 2023; Journée et al., 2023), flash flooding in urban areas and fast-responding catchments. Various studies have shown that state-of-the-art high-resolution hydrological model simulations are more responsive and perform better when fed with precipitation data at sub-daily intervals (e.g. Finnerty et al., 1997; Satish and Vasubandhu, 2013). This requires the development of high-resolution precipitation observation fields that can be used in both adjusting the model parameters and validating the quality of hydrological forecasts. Furthermore, gridded precipitation datasets have wide-ranging applications, such as providing the inputs to national hydrological modeling (Bell et al., 2007), the assessment of historical climate and its variability (e.g. Blenkinsop et al., 2008; Becker et al., 2013; Yu et al., 2016) and the assessment of reanalysis and downscaled climate model products (Isotta et al., 2015), as mentioned in Lewis et al. (2018). The main purpose of the current work is to develop such a gridded hourly precipitation dataset on a very long period using an analog method.

Lorenz made use of the concept of analogs in 1969 (Lorenz, 1969) to investigate the predictability limits of the atmosphere, and since then, the analog method has been widely used in many studies on various topics: purely data-driven forecasting (e.g. Toth, 1989; Yiou, 2014; Yiou and Déandréis, 2019), predictability analysis (e.g. Toth, 1991; Xavier and Goswami, 2007; Li and Ding, 2011), data assimilation (e.g. Lguensat et al., 2017), downscaling (e.g. Ghilain et al., 2021; Rozoff and Alessandrini, 2022), causality analysis (e.g. Sugihara et al., 2012; Vannitsem and Ekelmans, 2018; Huang et al., 2020), or even extracting dynamical properties of attractors (Foresti et al., 2024), among others.

The analog method consists in finding similarities within a collection of events, and to exploit these similarities for various purposes, in particular for emulating new events. These similarities are generally measured based on specific distances between the events under consideration. In atmospheric and climate dynamics, it often consists in measuring Euclidean distances between specific fields.

For downscaling purposes, it is hypothesized that local weather situations are influenced by large-scale synoptic-scale predictors. Essentially, if two synoptic situations are similar, they are likely to result in similar local effects. By using well-chosen atmospheric predictors, one can identify situations, say for a specific day, with the most comparable atmospheric conditions to a specific target day in order to predict the local properties of interest, such as for instance high resolution daily or hourly precipitation.

To find analogs, the synoptic weather pattern of the target day is analyzed, considering selected predictors. The analog technique searches for days that have similar atmospheric conditions within the time frame when the high resolution targeted local properties are available, without making assumptions about the probability distributions of the predictors (Raynaud et al., 2020). It is important to note that this approach relies on the assumption that similar atmospheric conditions will result in similar precipitation patterns. While this may not always hold true, this approach provides a useful starting point for creating a high-resolution gridded dataset. We will use this approach for inferring high-resolution hourly precipitation fields over Belgium for about the past 80 years using RADCLIM, a combined gauge-radar climatological dataset at a spatial resolution of 1x1 km<sup>2</sup> and at an hourly time scale developed by the Royal Meteorological Institute of Belgium (RMI). Figure 1 provides a visual overview of the method used. Section 2 will give an overview of the data used and the analog technique will also be explained. An analysis of the analog days found based on the selected predictor set will be presented in Section 3, together with the high-resolution hourly precipitation dataset. Section 4 gives a description of the organization of the dataset. The discussion and main conclusions are outlined in Section 5.

**Figure 1.** Overview figure illustrating the process of downscaling and validation of precipitation fields based on the analog method. The procedure in a nutshell: (i) selection of the analog days within the period covered by the radar based on large-scale fields of ERA5 to a specific target day in the past; (ii) association of the hourly precipitation fields of the analog days to the specific target in the past; (iii) construction of the best estimate of the precipitation field, either by using a single analog or a combination of them.

## 65 2 The data and the studied area

The downscaling analog-based approach for constructing high resolution precipitation dataset uses several datasets: (i) a large-scale database used for the identification of the analogs (ERA5); (ii) the high-resolution fields used as the downscaled proxy (Radar data); and (iii) the independent observations used to validate the downscaled fields (Automatic Weather Station, AWS, data from RMI). These three datasets are first described, together with the domain over which the analysis is performed.

## 70 2.1 The datasets

## 2.1.1 ERA5

This study used atmospheric reanalysis, specifically the ERA5 reanalysis on pressure levels from 1940 to the present (Hersbach et al., 2020), to identify analog days based on atmospheric conditions. ERA5 is the fifth generation European Centre for Medium-Range Weather Forecasts (ECMWF) reanalysis for global climate and weather over the past eight decades. The atmospheric parameters data is available at a spatial resolution of 0.25 degrees, enabling the identification of analog synoptic days.

Since ERA5 reanalysis data is only available from 1940, it is only possible to build a high-resolution precipitation series based on analogs from 1940. Since radar data is available from 2017, the emulated dataset is built until 2016. Three pressure level fields from the reanalysis are used: Geopotential (z), Temperature (t) and Relative humidity (r).

## 80 2.1.2 Radar data

The RADCLIM product is an offline counterpart to the real-time RADQPE (RADar Quantitative Precipitation Estimation) system (Goudenhoofdt and Delobbe, 2016). It reconstructs historical precipitation datasets using the same core methodology as RADQPE, but adapts it for retrospective analysis by incorporating missing radar or rain gauge measurements that were unavailable during real-time operations. This approach addresses the gaps and uncertainties inherent to instantaneous processing, resulting in a more complete and reliable dataset suited for climatological analyses.

RADCLIM provides high resolution radar-based quantitative precipitation estimation for Belgium and its surroundings. The RADCLIM product is obtained after a careful processing of the weather radar measurements and a merging with rain gauge measurements. This combination provides a detailed and accurate picture of precipitation distribution in time and space.

Data is collected through a network of weather radars and rain gauges. RADCLIM incorporates 3D reflectivity data from a network of four C-band dual-polarization weather radars: Jabbeke and Wideumont, owned and operated by RMI; Helchteren, managed by the Flanders Environment Agency (VMM); and Avesnois, operated by Meteo-France. The rain gauge data come from automatic networks of the Flemish Environment Agency (VMM), the Service Public de Wallonie (SPW), the Hydraulic Laboratory (WL), the Royal Meteorological Institute (RMI) and from manual rain gauges network CLIM of RMI. Both automatic and manual quality checks are carried out to ensure data reliability.

Radar reflectivity measurements are converted into rain rates. To get from raw radar reflections to reliable precipitation estimates, several processing steps are carried out. To mitigate non-meteorological echoes (e.g., ground clutter, wind farms), satellite cloudiness comparisons, vertical reflectivity profile analyses, and spatial texture detection are employed.

The data from the different radars are then merged into a precipitation composite. Precipitation accumulations are calculated over 5-minute and 1-hour intervals using optical flow techniques. These data are combined with rain gauge observations every 5 minutes via a statistical interpolation method based on the hypothesis of Gaussian process. This creates a correction factor, which is then also applied to the 5-minute accumulations. The final composite product has a spatial resolution of 1 km and uses the Belgian Lambert 2008 projection. Coverage extends from 0.3°W to 9.7°E and from 47.4°N to 53.7°N. Precipitation accumulations for 5 minutes and 1 hour are available based on UTC time and are provided in standard formats such as GeoTIFF and HDF5. An overview of the approach to combine the estimated precipitation fields with the Radar and the measurement network is presented in the Appendix of Journée et al. (2023). The quality of the precipitation fields generated by the method is also discussed in Journée et al. (2023) on the July 2021 flood event over Belgium.

The available archive of radar images used to build the RADCLIM dataset spans six years from 2017 to 2022. The hourly precipitation fields are available within this time frame. The effectiveness of finding suitable analogs for a specific target day is directly proportional to the size of the candidate analog database. The analog technique will only be able to emulate precipitation that were already recorded over the six-year period, which could of course be restrictive in particular for extreme events.

# 2.1.3 Rain gauge data (AWS)

100

105

110

125

Rain gauge data from the RMI's Automatic Weather Stations (AWS) have been available since 2003/2004. Prior to that, only 8h–8h precipitation data were recorded, which are not suitable when matching days based on 0h–0h accumulations. RMI's weather stations are part of a global network that essentially records ground-based observations of temperature, humidity, air pressure, wind and precipitation, among other things. The RMI's weather stations are fully automated and are providing measurements every 10 minutes. Precipitation is measured by weighing rain. Based on the data from an AWS station, hourly accumulations can be provided.

The 11 stations whose measurements were used were selected from across the country for optimal coverage, as illustrated in Figure 2. Furthermore, the stations were in operation during the case study year of 2006 and also from 2017 to 2022, the years for which radar data is available.

# 2.2 Study area

The research was carried out across a rectangular region covering Belgium, extending slightly wider to the west and north to encompass the dynamics over the North Sea. The topography changes from a flat sea and polder landscape to the low mountains of the Ardennes. The region considered to determine analog days extends from 1.5 to 6.5 longitude and 49.4 to 52.4 latitude, with an approximate area of 117 000 km<sup>2</sup>. With the spatial resolution of the ERA5 reanalysis, 273 grid points fill the rectangle as depicted in Figure 2.

**Figure 2.** The rectangle represented by dots covers Belgium and its near surroundings where the dots represent the grid points for which data are available in our selection of the ERA5 reanalysis. RMI stations across Belgium that provided observational precipitation data (Beitem, Zeebrugge, Melle, Ukkel, Dourbes, Ernage, Retie, Humain, Diepenbeek, Buzenol and Mont Rigi) are presented by the orange dots.

The radar composite cover an area from 0.3W to 9.7E in longitude and from 47.4N to 53.7N latitude.

# 3 Analog technique

140

130 The analog technique used in this study is based on the principle that local weather conditions are influenced by large-scale atmospheric patterns. According to the weather analog hypothesis, if two synoptic-scale situations are similar, they are likely to produce similar local weather outcomes. By comparing a given target day with past days that had similar atmospheric conditions, we aim to estimate local variables, such as precipitation. This approach assumes that similar large-scale atmospheric conditions lead to similar local precipitation patterns. While this assumption may not always hold perfectly, it provides a solid foundation for developing a high-resolution gridded precipitation dataset.

To identify analog days, we tested various combinations of atmospheric variables and time steps. The selection of predictors and time steps was guided by insights from previous research on analog methods. These previous studies have consistently shown that geopotential height is among the most effective predictors for identifying analog days, which is why it was prioritized from the outset (Horton et al., 2017; Horton, 2019). The results confirmed that sets including geopotential height consistently outperformed those without it. Due to the computational burden of selecting analogs and validation when testing multiple variables across many pressure levels and time steps, only a limited selection of predictor sets was addressed in line with the

previous literature. Therefore, we made informed, targeted selections to ensure feasibility, guided by a combination of trial-and-error and the findings from previous studies. Several combinations of fields and times of day were tested, and we ultimately arrived at a scheme that provides good results. However, we acknowledge that other combinations might perform even better, and further work is needed to optimize the method. These improvements are planned for the next version of the dataset. The identification of best analog days is therefore based on the following combination of predictors:

- (1) Geopotential height at 850 hPa at 0 and 12 UT (PRES012)
- (2) Relative humidity at 850 hPa at 9, 12 and 18 UT (RELHUM)
- (3) Temperature at 850 hPa at 12 UT (TEMP12)

145

- 150 (4) Geopotential height at 850 hPa at 6 UT, at 500 hPa at 12 UT and at 700 hPa at 24 UT (TWS3)
  - (5) (4) + Geopotential height at 1000 hPa at 12 UT (TWS4)

Furthermore to account for seasonality, candidate analog day searches are conducted within a 120-day window around the target day (60 days before and 60 days after). For example, candidate analog days for 19 March are selected within a window of days from 18 January till 18 May.

When concerning geopotential height, analogs are selected according to the Teweles–Wobus score (TWS) developed in Teweles and Wobus (1954). This score has been found to lead to higher performances than more classical Euclidean and Malahanobis distances (e.g. Kendall and Stuart, 1983; Guilbaud and Obled, 1998; Wetterhall et al., 2005; Raynaud et al., 2020). It quantifies the similarity between two geopotential fields comparing their spatial gradients and thus an analogy of the atmospheric circulation instead of considering the actual values at grid points (Horton, 2019). In this way we can select dates that have the most similar spatial patterns in terms of atmospheric velocity fields at several pressure levels (Raynaud et al., 2020). Analogs selected using predictive variables such as relative humidity and temperature are assessed based on the Root Mean Square Error (RMSE) metric.

# 3.1 Validation of the most appropriate method

The performances of the various indicator sets were evaluated by comparing the precipitation fields for the target day with those of the corresponding analog day, which was characterized by radar composite data collected at the same locations as the 11 measurement points. On the target day (in this case, a day in 2006), 24-hour precipitation measurements were available from 11 AWS stations. Once an analog day was identified using a given predictor set, the radar-based precipitation field for that analog day was used for comparison. Because radar grid points do not align exactly with the AWS station coordinates, we identified the four nearest radar grid points to each station. The mean precipitation from these four surrounding grid points was then used as the estimated 24-hour precipitation at that AWS location for the analog day. The value found as a result then counts as the amount of fallen precipitation at that particular location according to the radar data. The RMSE (Root Mean Square Error) was used to quantify the difference between two precipitation fields (each consisting of 11 point measurements) by considering the

corresponding grid points. This means that for every day, the analog precipitation field and the historical precipitation field are compared point by point and for every pair of fields the following measure was calculated:  $\sqrt{\frac{1}{n_s}\sum_s(p_{rs}-p_{as})^2}$ , with  $p_{rs}$  the precipitation amount in mm/24h on the historical day at location s,  $p_{as}$  the precipitation amount in mm/24h on the analog day at location s and s the number of stations. This measure is calculated for every single day and was then averaged over months, seasons and the whole year to make the comparison between the different methods more convenient. This average RMSE expressed in mm/24h as unit is displayed on the s-axis. Lower mean RMSE values indicate a smaller degree of difference between the precipitation fields which signifies higher accuracy. This metric is used for evaluating the performance of different selection methods.




**Figure 3.** Monthy (a), seasonly (b) and yearly (c) performance of the different predictor sets. The five methods under consideration were assessed on a monthly, seasonal, and annual basis. For each predictor set, the mean RMSE (in mm/24h) was computed for each time period (month, season, and year). The average of these summary statistics serves as an indicator of the overall performance of each predictor set. As showed in all three figures, it is clear at a glance that the TWS3 and TWS4 methods exhibit superior performance compared to RELHUM and TEMP12.

As shown in Figure 3, the options relying on geopotential height only produce the most accurate results when considering the best analog day. In the figure, the average root mean squared error (RMSE) for the various predictor sets concerning the best analog days is displayed on the y-axis as a function of time unit (months, seasons, years). A lower RMSE signifies superior average performance of the predictor set, thereby indicating better average first analogs. A significant difference in mean RMSE was observed among the five selection methods (p = 0.000561). A post-hoc analysis revealed that the 4TWS predictor set performed notably better (significant lower mean RMSE) than TEMP 12 (p = 0.00261) and RELHUM (p = 0.03603). There were no significant differences in mean RMSE between TWS4 and TWS3 and between TWS4 and PRES012. Based on these results and based on the choices made in the literature mentioned above, we will determine analogs based on this set of 4 geopotential height-related predictors (TWS4) for follow-up and analog dataset building.

#### 190 3.2 Validation of the selected predictor set (TWS4)

In the preceding section, the TWS4 predictor set was identified as the most effective. This section examines the top daily analogs discovered using this method and evaluates their statistical properties at daily and hourly time scales for 2006. The year 2006 was selected as the case study year because from that moment onward, a considerable number of automatic weather stations became available, and the year was relatively wet. While more 'average' years such as 2007, 2012 or 2014 could have been considered, our focus on evaluating precipitation fields led us to prefer a relatively wet year.

We also examined in details additional years after 2006. Overview figures for evolution of geopotential height for 2011 (drier year) and 2012 (wetter year) are added in the supplement to show the robustness of the results (see Figures S1, S2 and S3, and also the figures attached below). We found that the evolution of geopotential height remains consistent between the real days and the analog days identified using the 4TWS predictor set. The overall statistical properties of weather patterns are preserved across the different years, suggesting that the method reliably captures the key dynamical features and precipitation patterns. By extension, a 10-year period (1990 – 1999) will also be looked at more closely by considering the best analog for each target

Just as Uccle serves as the national reference station in Belgium for monthly, seasonal, and yearly climatological reports, as well as for trend analysis, it was also chosen as the reference and evaluation location in this study.

day as well as an ensemble of analog days with a variable number of members.

#### 205 3.2.1 Statistics on geopotential height


Figure 4 shows the temporal evolution of the geopotential height at 12 UT at the 500hPa pressure level in Uccle on consecutive best analog days for 2006. The overall trend appears to be consistent in both scenarios, with a minimum at the end of winter and a maximum at the end of summer. However, upon closer examination, it is found that the geopotential height on real days (RD) exhibits less extreme fluctuations compared to the analog days (AD) (range AD: 5099 m – 5961 m; range RD: 5167 m – 5894 m).

**Figure 4.** The temporal evolution in geopotential height at the 500 hPa pressure level at 12 UT in Uccle in 2006 (panel a); the corresponding data on the best analog days in 2006 (panel b); differences in geopotential height on analog days, real days and random days (panel c). The yellow curves in the first two plots aid the eye in seeing patterns in the presence of overplotting by fitting a smooth curve to the data.

The differences in geopotential height for consecutive days on analog days appear to show larger jumps and less persistence compared to real days. The term 'difference' refers to the daily consecutive changes in geopotential height. Specifically, for the entire year, the change in geopotential height between 12:00 UTC on day x and 12:00 UTC on day x+1 is calculated. This provides an indication of the temporal consistency of the geopotential height on consecutive analog days. A closer analysis of these differences between the 365 open successive days as shown in the third panel of Figure 4, especially their standard deviations, confirms this suspect (SD AD = 143; SD RD = 68; SD Random = 218). For reference, the histogram also shows the distribution of geopotential height differences between consecutive days in 2006 at 500 hPa pressure level for Uccle, for randomly generated analog days.

The distribution of differences shows a larger spread (and thus more extreme differences in geopotential height between consecutive days) for the analog days than for the real days. When comparing the 364 differences between randomly generated analog days, it is evident that the actual analogs have a better preservation of dispersion and persistence (F-test on differences of random days and analog days gives p-value of  $8.882 \cdot 10^{-16}$ ). This indicates that selecting the best analogs based on the TWS4 predictor set seems appropriate.


Nevertheless, the low-frequency variability visible in panel (a) with long periods of low or high values seems less well represented. A possible reason is the limited period within which the analogs are selected (2017-2022).

**Figure 5.** Monthly statistics of geopotential height at 500 hPa pressure level in Uccle at 12 UT in 2006. Top left to bottom right: (a) mean, (b) minimum, (c) maximum and (d) standard deviation. The general pattern between real and analog days seems similar across all four summary measures.

In Figure 5 and Figure 6, the graphs demonstrate the trend of various descriptive statistics of geopotential height at best analog days when averaged over months and seasons. All the figures show that the analog days approximate the pattern of the real days quite well. This further gives confidence in the analogs chosen through the TWS4 predictor set as being a valid approximate representation of reality.

**Figure 6.** Seasonal statistics of geopotential height at 500 hPa in Uccle at 12UT in 2006. Top left to bottom right: (a) mean, (b) minimum, (c) maximum and (d) standard deviation. The general pattern seems similar in the different cases.

Also in the ten-year period from 1990 to 1999, a similar pattern can be seen in the temporal evolution of the geopotential height at 12UT at the 500 hPa pressure level in Uccle between the target period and the consecutive best analog days (figure not shown).

## 3.2.2 Statistics on precipitation fields



To assess the quality of daily precipitation estimation, observed 24-hour precipitation values at RMI stations on real days were compared with the estimated precipitation from the best radar analog days. As explained before, the precipitation amount observed by the radar at a specific location is found by averaging the 4 closest grid values.

An assessment of the quality of analog days was checked by looking at the daily average per month, as well as comparing the monthly standard deviation between the target day and the best analog day at different locations. As an example, Figure 7 for Uccle is shown here, but a similar picture is seen at other locations considered. We see similar trends between real and analog days, although the period we are looking at here is short, this image suggests that the approach in identifying analogs for 24-hour precipitation accumulation, is approriate.

Figure 7. Evolution of (a) daily mean per month and (b) monthly standard deviation of 24h precipitation accumulation in Uccle in 2006.

## 3.2.3 ROC curve


The Receiver Operating Characteristic (ROC) curve serves as a widely used diagnostic tool in forecast verification, with the Area Under the ROC Curve (AUC) used as a verification metric for evaluating the discrimination ability of a forecast (Ben Bouallègue and Richardson, 2022). Specifically, for precipitation forecasts, the ROC curve provides a valuable tool for assessing the performance of models across varying thresholds. This curve illustrates the relationship between the true positive rate (sensitivity) and the false positive rate (1-specificity) at multiple threshold levels (Ben Bouallègue and Richardson, 2022). We will use ROC curve analysis to assess the performance of predictive models based on the TWS4 predictor set, using various summary statistics and a variable number of analogs corresponding to each target day (i.e. variation in the ensemble size). It is

important to note that the analysis was perfored specifically for Uccle for 2006 on the one hand and for the decade 1990-1999 on the other hand.

Figure 8 shows the ROC curves representing the performance of each summary statistic model derived from this TWS4 selection approach at various classification thresholds, from 0 mm to higher amounts, for both 2006 (left panel) and the decade 1990 – 1999 (central panel). We compare three distinct models of 24-hour precipitation prediction, which rely on (1) the mean, (2) the median and (3) the maximum. These mean, median and maximum were calculated from the 24-hour precipitation of a 10 ensemble member, represented by the best 10 analog days for every target day.




The selected threshold values are location-specific and are determined based on percentiles derived from the actual distribution of 24-hour precipitation accumulation for the specific location, here Uccle. From top right to bottom left, the threshold values used in the ROC curves in Figure 8 are: 0 mm, 0.1 mm, 0.2 mm, 0.54 mm, 0.8 mm, 1.78 mm, 3.6 mm, 4.74 mm, 6.66 mm, 9.98 mm, 27.3 mm.

**Figure 8.** From left to right: (a) ROC curve for Uccle for 2006 for models based on different summary statistics using the 10 best analogs; (b) ROC curve for Uccle for the period 1990 - 1999 for models based on different summary statistics using the 10 best analogs; (c) The evolution of AUC values in relation to the increasing number of analog days per target day in 1990-1999, using the median as the predictive measure.

The AUC (Wilks, 2011) provides a scalar representation of each model's performance: an AUC of 1 corresponds to perfect classification, while an AUC of 0.5 indicates no discriminative power. This makes it easy to compare different models or classifiers quantitatively. In particular, the model using the median of the 24-hour precipitation accumulation from the 10-member analog ensemble yields a slightly higher AUC value, indicating its superior performance compared to the other models. This observation opens the question of how the median evolves as the number of ensemble members increases.

Panel (c) of Figure 8 shows the performance of the median calculated from 2 to 27 analogs. The result indicates that the AUC improves with the addition of more members; however, we notice a plateau in performance starting at  $\pm$  23 analogs. This

suggests that while the model benefits from the inclusion of additional analogs, there are diminishing returns beyond this point. Consequently, it appears that incorporating 25 analogs into the model is optimal to achieve the best performance.

Given the wealth of information captured in the ROC curves, an alternative representation offers complementary insights. Figures 9, 10 and 11 present separate ROC curves for three precipitation thresholds, plotting the hit rate (HR) versus the false alarm rate (FAR) based on the number of ensemble members exceeding each threshold. This differentiation between light (1mm/24h), medium (5mm/24h) and heavy precipitation (15mm/24h) enables a nuanced analysis of the model's performance under varying precipitation conditions.



Figure 9 presents probabilistic forecasts generated from the 10-member analog ensemble for 2006 at Uccle – only 4 events in 2006 exceeded 15mm/24h, which is not much and why it is useful to consider a longer period of time. Figure 10 also considers an ensemble with 10 members, but this time with statistics over the longer period 1990 - 1999. Figure 11 considers a 25-member ensemble for this 10-year period. A closer look at the ROC curves reveals that our approach has a high discriminative ability for the 1 mm threshold, and despite the fact that this performance deteriorates as the threshold increases, it still maintains good performance. The discriminative ability associated with a threshold value of 15 mm exhibits an approximate increase of 20% when using a 25-member ensemble compared to a 10-member ensemble. This enhancement is remarkably greater than that observed for the 1- and 5-mm thresholds, indicating that models designed to perform well under conditions of elevated precipitation benefit from the inclusion of a larger ensemble size, as these events occur less frequently.

**Figure 9.** Examples of ROC curves for common and rare events. ROC curves and corresponding AUCs for precipitation forecasts with event thresholds: 1 mm, 5 mm, and 15 mm per 24h (2006, 10-member ensemble).

Figure 10. Examples of ROC curves for common and rare events during the 10-year period 1990 – 2000 for 10 analogs. ROC curves and corresponding AUCs for precipitation forecasts with event thresholds: 1 mm, 5 mm, and 15 mm per 24h (1990 – 1999, 10-member ensemble).

Figure 11. Examples of ROC curves for common and rare events during the 10-year period 1990 – 2000 for 25 analogs. ROC curves and corresponding AUCs for precipitation forecasts with event thresholds: 1 mm, 5 mm, and 15 mm per 24h (1990 – 1999, 25-member ensemble).

Figure 12 further illustrates the relationship between the AUC and the number of ensemble members across three specified precipitation thresholds. For scenarios focusing only on low precipitation events, say 1 mm/24h, a 15-member ensemble is


adequate, as increasing the ensemble size does not significantly enhance model performance. In contrast, for higher threshold rainfall scenarios, say 5 mm/24 hours, a 22-member ensemble appears to provide a relatively reliable model. Furthermore, the performance of the model for rainfall events with 15 mm/24h or more continues to improve with an increasing number of ensemble members, suggesting that a performance ceiling has not yet been reached. This indicates the necessity for an ensemble including more than 25 analogs to achieve optimal performance for more heavy precipitation events. This important result indicates that more analogs are needed for getting good information on the extreme precipitation cases, and this implies that the number of analogs to choose could be case-specific.

**Figure 12.** Evolution of AUC values based on the median with increasing number of ensemble members for the three considered event thresholds 1 mm, 5 mm and 15 mm per 24h for the 10-year period 1990 – 1999.

#### 3.3 Distribution of precipitation amounts based on the selection of a set of analogs




We have shown that it can be useful to optimize the number of analogs by using summary statistics to provide an estimate for the daily precipitation accumulation on a given day in Uccle. The distribution of precipitation on these best analog days, combined with the actual measured amount of precipitation, can also provide an important insight into where the actual value lies, given the distribution.

For Uccle, Beitem and Buzenol for example, this information is available as we have measurements from the AWS. An example of the distribution of 24-hour precipitation accumulation for the top 10 analog days for each day of the month, together with the actual values for January 1995 (month known for its high precipitation levels), is illustrated in Figure 13. This figure shows how the precipitation from radar data on analog days allows us to get an idea of the distribution of the expected precipitation values. The observed values frequently fall within the range of values represented by the analog days.

An advantage of the high-resolution dataset for precipitation is that this information is not limited to locations where an AWS is available. For any location in Belgium, by selecting the nearest grid point, one can find an analog day precipitation distribution. This increases the applicability of our model and makes it possible to generate precipitation estimations for areas that might otherwise lack sufficient data. An example of this is presented in Figure S4 of the Supplemental document where two locations, Diksmuide and Jalhay, have been selected in which distribution data can be inferred using analogs, despite the absence of monitoring points and corresponding observations.

**Figure 13.** Daily precipitation accumulation patterns (distributions) for January 1995 using the ten most similar analog days using radar values indicated by the blue dots. The daily observations obtained from the pluviometer in (a) Uccle, (b) Beitem and (c) Buzenol indicated by the purple markers. The yellow crosses represent the median precipitation value for each day based on these 10 best analog days.

# 3.4 Autocorrelation



In the analysis of consecutive best analog days, it is important to ensure that the underlying patterns observed in the hourly data exhibit similarity to those identified in actual data. When evaluating the autocorrelation within the context of analog days, the primary focus is on the correlation of the time series with itself at a lag of one hour. These lags are particularly important as they capture the transitions between successive analog days, which is imposed by the procedure we used to select the successive analogs.

The autocorrelation of the analog year 2006 was checked across five distinct locations in Belgium, as shown in Figure 14. The autocorrelation analysis for Uccle was performed for the median of the 10 best analog days of every target day. For the other four locations the hourly precipitation accumulations on the best analog day were considered for autocorrelation analysis. For three of these locations, the autocorrelation coefficient of the analog time series at lag 1 exceeded that of the observed time series. In contrast, for the locations of Uccle and Ernage, the autocorrelation at lag 1 was marginally lower; however, the observed decrease (4 and 15% respectively) does not raise concerns regarding the autocorrelation characteristics of the time series in these cases. Thus, there is no evidence of any autocorrelation issues in the data.

**Figure 14.** (a) Autocorrelation Function (ACF) for hourly data from Uccle in 2006, comparing real hourly AWS data with median hourly radar data based on the 10 best analogs in Uccle in 2006. (b) ACF for hourly data from Beitem, Mont Rigi, Ernage en Diepenbeek in 2006 on real and best analog days using lag period of 5.

# 4 Data availability



The dataset is relying on the *Zarr* archiving format (Miles et al., 2020) and is composed of two such archives. The first archive is a database and contains a matching set of 25 best analog days<sup>1</sup> for each target day from 1940 to 2016. The second archive contains the median hourly precipitation field of the 25 member ensemble (set of 25 best analogs for one day in the past) for every day from 1940 to 2016, calculated from the climatological RADCLIM radar product. The Zarr archives have been constructed using the *xarray* Python library (Hoyer and Joseph, 2017), and are licensed under the Creative Commons Attribution 4.0 International license (CC BY 4.0). The dataset is available on the Zenodo platform (https://doi.org/10.5281/zenodo.14965710) (Debrie et al., 2025). A notebook showing how to load and explore the dataset in Python is available at https://github.com/ElkeDebrie/RADCLIM-Analogs.

Figure 15 presents an example precipitation field from the published dataset for 27.01.2001. The left panel shows the observed daily precipitation accumulation for that date (grid based), while the right panel displays the daily accumulation (sum) of the median of the 25 best analogues for that day. The published dataset includes such median-based precipitation fields for each day between 1940 and 2016.

<sup>&</sup>lt;sup>1</sup>The database only contains the dates information, not the fields.

Figure 15. Comparison of observed daily precipitation accumulation and the median accumulation derived from the 25 best analog days.

For days with medium or heavy precipitation, it may be interesting to look at the ensemble mean precipitation instead of the median. If many analog days have zero precipitation, using the median of their precipitation values can be misleading: it will tend to be zero or very low. This means higher precipitation amounts won't be well represented, even if a few analog days had significant rainfall. In such cases, using the mean instead of the median can help. The mean takes into account all values, including the higher ones, so it can better reflect the presence of heavier precipitation within the set of analog days even if most of them were dry. The dataset with mean hourly precipitation is not made available through Zenodo due to its size, but if desired, this data can be requested from RMI Communication and Marketing service (marketing@meteo.be). Note that the RADCLIM data are not open but, as for the analogs mean, access to it can be requested for scientific purposes to RMI.

## 5 Conclusions





Spatially and temporally high-resolution climate data have become increasingly important. The main purpose of the current work was to develop a gridded hourly precipitation dataset over a very long period of 77 years for Belgium, featuring a resolution of 1 km, using an analog method. We used ERA5 reanalysis data at pressure levels relevant to large-scale atmospheric processes from ECMWF and appropriate metrics to determine analog days. Radar data and AWS data were used to evaluate the effectiveness of the best analogs of each predictor set. The most effective predictor set turned out to be those based on the four geopotential height pressure levels at varying timestamps and was then chosen for generating analogs while accounting for seasonal variations by restricting our selection to past dates within the same 4-month temporal windows.

Statistical evaluation of the selected predictor set indicates that the geopotential height patterns observed on actual consecutive days closely resemble those on consecutive analog days, with only minor discrepancies. Furthermore, autocorrelation analysis

shows no substantial differences between true consecutive hours and analog consecutive hours, particularly during transitions between two consecutive analog days.

ROC curves for the summary statistics (mean, median and maximum of 10 best analog days) of the 24-hour precipitation accumulation were used as a tool for evaluation of performance of the preferred model based on the predictor set with four levels of geopotential height. The analysis was performed for Uccle, specifically for 2006 and for the decade 1990-1999.

The analysis reveals that the median-based model yields the best ROC curves and the highest AUC, suggesting superior effectiveness among the tested models. As the number of analog members is increased, AUC values improve, reaching a plateau around 23 members, indicating diminishing returns past this point. Additional analysis on specific precipitation thresholds have revealed stronger model performance for low thresholds (1 mm/24h) but decreased effectiveness for higher thresholds (5 mm/24h and 15 mm/24h).





This first dataset of houly median precipitation values calculated based on the 25 best analogs at a space scale of 1 km for the period of 1940 to 2016 is now available for use on Zenodo (Debrie et al., 2025) (https://doi.org/10.5281/zenodo.14965710).

The full dataset with 25 analogs for every target day can be requested from the RMI. This dataset has a lot of potential as it can be used for a variety of purposes. To analyze the evolution of precipitation along the years under climate change and to calibrate and/or evaluate hydrological models, among others. A particularly interesting application is the use of this dataset to train machine learning models for both deterministic and probabilistic forecasting or downscaling. Short-term precipitation forecasting based on Machine Learning will be explored in the near future.

We acknowledge that the training dataset, which covers only a six-year period, is relatively short. It is a sophisticated merging of the radar datasets and the rain gauges data (Journée et al., 2023). Moreover, a coherent coverage with the same radars started only in 2017. Efforts are currently ongoing at the Royal Meteorological Institute to extend this dataset for the following years and also to extract the best information from previous years.

The risk of misrepresenting climate change is an important problem that could affect the results. The question is twofold: On one hand the change of precipitation intensity and on the other hand, the change of circulation patterns. For the precipitation intensity, climate change has currently a very limited impact over Belgium as reflected by the long time series covering Belgium (RMI, a). In the Uccle time series of 1-hour or 24-hour extremes, there is no obvious trend, except some long term modulation probably associated with low-frequency variability that still needs to be uncovered. In the recent attribution study of the extreme July 2021 case which affected the East of Belgium it was shown that there is no local trends of one-day or two-day extreme precipitation accumulations, except the extreme July event, which appears like a black swan (Tradowsky et al, 2023). We therefore suspect that climate change does not affect much (currently) the extreme intensities. Therefore if there are changes in precipitation it should mainly be related to changes in circulation as illustrated on the website of the Royal Meteorological Institute (RMI (b), RMI (c)). The analog approach should partly take that into account as the reconstruction of precipitation is done using analogs of weather situations (regimes) that were experienced in the past. So if the circulation regimes (but not necessarily their frequency) were the same in the training dataset and in the past, the method should appropriately take the change of frequency into account. As far as we can experience, there is no major disruption on the weather patterns (large scale pressure fields) yet that would imply that the analog approach cannot provide useful information on past weather

situations. This question of change of circulation regimes is however important to address in the future due to the rapid changes experienced by our climate. Such an analysis will then be used to support further developments of the analog dataset.




There is room for improving the model used to identify analog days. One way for improvement involves exploring more sophisticated predictor sets, including a wider array of variables at various pressure levels. Currently, when accounting for multiple good analog days per target day, only summary measures of 24-hour accumulations on these days were considered. Further research could reveal whether assigning well-chosen weights to these accumulations would provide a better performing model. In addition, the median was found to be the best overall summary measure for precipitation under the considered ones 395 and therefore published in a dataset, however on days with more than 1mm of precipitation the mean is the better option. As 66% of the days in the period 1940 till 2016 had less than 1mm of precipitation at the considered locations and as with the median a lot zeros were captured.

Author contributions. SV proposed the development of an analog-based downscaling precipitation dataset for Belgium. ED and SV designed the experiments. ED performed the technical developments. ED and JD developed the zarr dataset. ED wrote the initial draft of the manuscript. JD and SV contributed to the writing of the final manuscript.

Competing interests. The contact author has declared that neither him nor his co-authors have any competing interests.

Acknowledgements. We would like to thank Edouard Goudenhoofdt for providing the RADCLIM data containing hourly high resolution radar-based quantitative precipitation estimation for Belgium and its surroundings. We would also like to thank Michel Journée for sharing his knowledge on the availability of precipitation data in the present and past and to supply information about the RMI Automatic Weather Station Network. Furthermore, we would like to thank Nicolas Ghilain for his willingness to engage in discussions about our project, Lesley De Cruz for sharing her critical perspective, insights and feedback regarding the ROC curves presented in this paper and Maryna Lukach and Dieter Poelman for sharing their expertise and reviewing the content related to RADCLIM. Finally, We would like to acknowledge the use of artificial intelligence tools in the preparation of this manuscript: AI was employed to optimize phrasing and enhance the clarity of our text, contributing to a more coherent presentation of the research findings.

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
