# Peer review of "Hourly precipitation fields at 1 km resolution over Belgium from 1940 to 2016 based on the analog technique"

_Earth System Science Data, 2025_

## Referee Comment (RC1)

Review for „**Hourly precipitation fields at 1 km resolution over Belgium from 1940 to 2016 based on the analog technique**" submitted to *Earth System Science Data* by Elke Debrie et al.

**General comments:**

This publication presents an analogue method to determine hourly rainfall patterns over Belgium for a 80-year period based on radar observation.

This method shows good potential for gridded precipitation fields over this period, but my main concern is that the training dataset is quite short. In addition, it is at the end of the time period (1940-now). Within this time period, climate change caused an increase of mean temperatures by about 1.5°C in central and Western Europe which also caused a change in circulation patterns. Please discuss this in the paper.

Why did you use 2006 as a target year for evaluation? It was a particular year with very different weather conditions (cold winter, hot and dry June/July, cold and wet August, very mild autumn). It would be interesting to compare the analogues for more than one year in section 3.2.1/3.2.2. Do the results (esp. about differences) change?

**Specific comments:**

Section 2.1.2: The RADCLIM product is based on a combination of radar data merged with ground-based precipitation observations? This is not entirely clearly written here.

Section 3, lines 108-112: How did you choose these parameters, especially concerning (4) – why did you use the geopotential at each level at different times of the day? Do you have a physical reasoning for this? Did you try other combinations?

Figure 3: Which unit does the "mean RMS distance" have? Precipitation (mm per day/hour)? Please explain the meaning of these numbers!

Figure 4, panel c: You write: "differences in geopotential height on analog days, real days and random days". What is the reference of the differences here? Is it compared to the smoothed yellow line?

**Technical comments:**

Line 201: I couldn't find the reference "Wil, 2019"

Figures 9-11: To me, it would make sense to merge these plots (9a+10a+11a in one plot with three lines, and correspondingly for the others)

Also Figures 12 a-c should be merged into one plot

---

## Author Comment (AC1)

**Review 1**

**Review for „Hourly precipitation fields at 1 km resolution over Belgium from 1940 to 2016 based on the analog technique" submitted to Earth System Science Data by Elke Debrie et al.**

**General comments:**
This publication presents an analogue method to determine hourly rainfall patterns over Belgium for a 80-year period based on radar observation.

This method shows good potential for gridded precipitation fields over this period, but my main concern is that the training dataset is quite short. In addition, it is at the end of the time period (1940-now). Within this time period, climate change caused an increase of mean temperatures by about 1.5°C in central and Western Europe which also caused a change in circulation patterns. Please discuss this in the paper.

Thank you very for drawing our attention to these important problems. Indeed climate change may affect the analog selection. On one hand, climate change may affect the intensity of precipitation, and on the other hand, the circulation patterns (their frequency or the emergence of new patterns) that lead to specific precipitation fields. Our analysis indeed assumes that the precipitation intensity is not strongly modified and that no new emerging circulation patterns are formed yet. Concerning the former, our time series suggests that there is no substantial trend in the intensity at short time scale (1 hour, 1 day). Concerning the latter, we did not see any report of the emergence of new patterns, but well of the possible change of frequency which is naturally taken into account by the selection of analogs, i.e. only analogs of the specific circulation pattern are in principle selected.

As you will see below in the text we underline the absence of substantial trends in extreme precipitation due to climate change, and we discuss the question of the modification of circulation patterns that could affect the frequency of events. Both points are now discussed in the text as follows in Section 5:

*"We acknowledge that the training dataset, which covers only a six-year period, is relatively short. It is a sophisticated merging of the radar datasets and the rain gauges data \citep{journee2023}. Moreover, a coherent coverage with the same radars started only in 2017. Efforts are currently ongoing at the Royal Meteorological Institute to extend this dataset for the following years and also to extract the best information from previous years.*

*The risk of misrepresenting climate change is an important problem that could affect the results. The question is twofold: On one hand the change of precipitation intensity and on the other hand, the change of circulation patterns. For the precipitation intensity, climate change has currently a very limited impact over Belgium as reflected by the long time series covering Belgium \citep{RMI1}. In the Uccle time series of 1-hour or 24-hour extremes, there is no obvious trend (as for temperature for instance), except some long term modulation probably associated with low-frequency variability that still needs to be uncovered. In the recent attribution study of the extreme July 2021 case which affected the East of Belgium it was*

*shown that there is no local trends of one-day or two-day precipitation accumulations, except the extreme July event, which appears like a black swan (Tradowsky et al, 2023). We therefore suspect that climate change does not affect much (currently) the extreme intensities. Therefore if there are changes in precipitation it should mainly be related to changes in circulation as illustrated on the website of the Royal Meteorological Institute (\cite{RMI2}, \cite{RMI3}). The analog approach should partly take that into account as the reconstruction of precipitation is done using analogs of weather situations (regimes) that were experienced in the past. So if the circulation regimes (but not necessarily their frequency) were the same in the training dataset and in the past, the method should appropriately take the change of frequency into account. As far as we can experience, there is no major disruption on the weather patterns (large scale pressure fields) yet that would imply that the analog approach cannot provide useful information on past weather situations. This question of change of circulation regimes is however important to address in the future due to the rapid changes experienced by our climate. Such an analysis will then be used to support further developments of the analog dataset."*

Why did you use 2006 as a target year for evaluation? It was a particular year with very different weather conditions (cold winter, hot and dry June/July, cold and wet August, very mild autumn). It would be interesting to compare the analogues for more than one year in section 3.2.1/3.2.2. Do the results (esp. about differences) change?

Thank you for raising this point. The results are in fact robust among the different years, and we selected 2006 as an example for displaying the results as the number of automatic stations have increased at that moment. Moreover this year was relatively wet. The following paragraphs are added to Section 3.2:

*"The year 2006 was selected as the case study year because from that moment onward, a considerable number of automatic weather stations became available, and the year was relatively wet. While more 'average' years such as 2007, 2012 or 2014 could have been considered, our focus on evaluating precipitation fields led us to prefer a relatively wet year.*

*We also examined in details additional years after 2006. Overview figures for evolution of geopotential height for 2011 (drier year) and 2012 (wetter year) are added in the supplement to show the robustness of the results (see Figures S1, S2 and S3, and also the figures attached below). We found that the evolution of geopotential height remains consistent between the real days and the analog days identified using the 4TWS predictor set. The overall statistical properties of weather patterns are preserved across the different years, suggesting that the method reliably captures the key dynamical features and precipitation patterns."*

**Specific comments:**

Section 2.1.2: The RADCLIM product is based on a combination of radar data merged with ground-based precipitation observations? This is not entirely clearly written here.

Thank you for raising this point. The text about RADCLIM has now been expanded as follows in Section 2.1.2:

*"The RADCLIM product is an offline counterpart to the real-time RADQPE (RADar Quantitative Precipitation Estimation) system \citep{ GenerationandVerificationofRainfallEstimatesfrom10YrVolumetricWeatherRadarMeasureme nts}. It reconstructs historical precipitation datasets using the same core methodology as RADQPE, but adapts it for retrospective analysis by incorporating missing radar or rain gauge measurements that were unavailable during real-time operations. This approach addresses the gaps and uncertainties inherent to instantaneous processing, resulting in a more complete and reliable dataset suited for climatological analyses.*

*RADCLIM provides high resolution radar-based quantitative precipitation estimation for Belgium and its surroundings. The RADCLIM product is obtained after a careful processing of the weather radar measurements and a merging with rain gauge measurements. This combination provides a detailed and accurate picture of precipitation distribution in time and space.*

*Data is collected through a network of weather radars and rain gauges. RADCLIM incorporates 3D reflectivity data from a network of four C-band dual-polarization weather radars: Jabbeke and Wideumont, owned and operated by RMI; Helchteren, managed by the Flanders Environment Agency (VMM); and Avesnois, operated by Meteo-France. The rain gauge data come from automatic networks of the Flemish Environment Agency (VMM), the Service Public de Wallonie (SPW), the Hydraulic Laboratory (WL), the Royal Meteorological Institute (RMI) and from manual rain gauges network CLIM of RMI. Both automatic and manual quality checks are carried out to ensure data reliability.*

*Radar reflectivity measurements are converted into rain rates. To get from raw radar reflections to reliable precipitation estimates, several processing steps are carried out. To mitigate non-meteorological echoes (e.g., ground clutter, wind farms), satellite cloudiness comparisons, vertical reflectivity profile analyses, and spatial texture detection are employed. The data from the different radars are then merged into a precipitation composite. Precipitation accumulations are calculated over 5-minute and 1-hour intervals using optical flow techniques. These data are combined with rain gauge observations every 5 minutes via a statistical interpolation method based on the hypothesis of Gaussian process. This creates a correction factor, which is then also applied to the 5-minute accumulations.*

*The final composite product has a spatial resolution of 1 km and uses the Belgian Lambert 2008 projection. Coverage extends from 0.3°W to 9.7°E and from 47.4°N to 53.7°N. Precipitation accumulations for 5 minutes and 1 hour are available based on UTC time and are provided in standard formats such as GeoTIFF and HDF5."*

Section 3, lines 108-112: How did you choose these parameters, especially concerning (4) – why did you use the geopotential at each level at different times of the day? Do you have a physical reasoning for this? Did you try other combinations?

Thank you for your feedback. To clarify this aspect, we changed the text in the beginning of Section 3 as follows:

*"The analog technique used in this study is based on the principle that local weather conditions are influenced by large-scale atmospheric patterns. According to the weather analog hypothesis, if two synoptic-scale situations are similar, they are likely to produce similar local weather outcomes. By comparing a given target day with past days that had similar atmospheric conditions, we aim to estimate local variables, such as precipitation. This approach assumes that similar large-scale atmospheric conditions lead to similar local precipitation patterns. While this assumption may not always hold perfectly, it provides a solid foundation for developing a high-resolution gridded precipitation dataset.*

*To identify analog days, we tested various combinations of atmospheric variables and time steps. The selection of predictors and time steps was guided by insights from previous research on analog methods. These previous studies have consistently shown that geopotential height is among the most effective predictors for identifying analog days, which is why it was prioritized from the outset \citep{horton2017, horton2019}. The results confirmed that sets including geopotential height consistently outperformed those without it.*

*Due to the computational burden of selecting analogs and validation when testing multiple variables across many pressure levels and time steps, only a limited selection of predictor sets was addressed in line with the previous literature. Therefore, we made informed, targeted selections to ensure feasibility, guided by a combination of trial-and-error and the findings from previous studies. Several combinations of fields and times of day were tested, and we ultimately arrived at a scheme that provides good results. However, we acknowledge that other combinations might perform even better, and further work is needed to optimize the method. These improvements are planned for the next version of the dataset."*

Figure 3: Which unit does the "mean RMS distance" have? Precipitation (mm per day/hour)? Please explain the meaning of these numbers!

The unit – mm/24h – is added to the plots and in the text. A more clear explanation about the meaning of the mean RMSE is now added to Section 3.1:

*"For every day, the analog precipitation field and the target precipitation field are compared and for every pair of fields the following measure was calculated: $\sqrt{\frac{1}{n_s}\sum_s(p_{rs} - p_{as})^2}$, with $p_r$ the precipitation amount in mm/24h on the real day, $p_a$ the precipitation amount in mm/24h on the analog day, $s$ the AWS and $n_s$ the number of stations. This measure is calculated for every single day and was then averaged over months, seasons and the whole year to make the comparison between the different methods more convenient. This average RMSE with mm/24h as unit is displayed on the y-axis."*

Figure 4, panel c: You write: "differences in geopotential height on analog days, real days and random days". What is the reference of the differences here? Is it compared to the smoothed yellow line?

The term 'differences' was explained more clearly in Section 3.2.1:

*"The term 'difference' refers to the daily consecutive changes in geopotential height. Specifically, for the entire year, the change in geopotential height between 12:00 UTC on*

*day x and 12:00 UTC on day x+1 is calculated. This provides an indication of the temporal consistency of the geopotential height on consecutive analog days."*

**Technical comments**
Line 201: I couldn't find the reference "Wil, 2019"

Thank you for pointing that out. The reference was not made well in the LaTeX document. It should be fine now.

Figures 9-11: To me, it would make sense to merge these plots (9a+10a+11a in one plot with three lines, and correspondingly for the others) Also Figures 12 a-c should be merged into one plot

Thank you very much for the suggestion. Now, panels a, b and c from Figures 9, 10, 11 and 12 were merged into one plot in Figure 9 till Figure 12 in Section 3.2.3

Additional figures added to the supplement:

[Figure]

Figure S1. The temporal evolution in geopotential height at the 500 hPa pressure level at 12 UT in Uccle in 2011 resp. 2012 (panel a); the corresponding data on the best analog days in 2011 resp. 2012 (panel b); differences in geopotential height on analog days, real days and random days in 2011 resp. 2012 (panel c). The yellow smoothed curves in the first two plots aid the eye in seeing patterns.

[Figure]

Figure S2. Monthly statistics of geopotential height at 500 hPa pressure level in Uccle at 12 UT in 2011 and 2012. Respectively mean, minimum, maximum and standard deviation are shown in the figures. Consistent with previous observations, the general pattern between real and analog days is comparable across all four summary measures.

[Figure]

Figure S3. Seasonly statistics of geopotential height at 500 hPa pressure level in Uccle at 12 UT in 2011 and 2012. Respectively mean, minimum, maximum and standard deviation are shown in the figures. The general pattern observed across all four summary measures shows a high degree of similarity between real and analog days, aligning with earlier results.

**References**

Tradowsky, J. et al, Attribution of the heavy rainfall events leading to the severe flooding in Western Europe during July 2021. *Climatic Change*, 176, 90, https://doi.org/10.1007/s10584-023-03502-7, 2023 .

---

## Author Comment (AC2)

**Review 2**

**Review of manuscript "Hourly precipitation fields at 1 km resolution over Belgium from 1940 to 2016 based on the analog technique"**

**General comments**

This manuscript presents a long-term precipitation data set for Belgium based on the analog method sampling from a radar data set of six years. A precipitation data set with an hourly temporal and 1 km spatial resolution for a time period of 77 years is impressive. In general, such a data set has potential and fits the scope of ESSD well. However, I agree with the concerns of the referee #1 that the sampling from a five-year radar data set is too short and not well representative. Additionally, the description methodology could be improved and the description of the archive data set is sparse.

- I agree with the concerns of the referee #1. The sampling from a six-year radar data set, which is at the end of the period 1940 to 2016 seems to be too short and misrepresenting taking the climate change into account. Please discuss this in your paper.

Thank you very for drawing our attention to these important problems. Indeed climate change may affect the analog selection. On one hand, climate change may affect the intensity of precipitation, and on the other hand, the circulation patterns (their frequency or the emergence of new patterns) that lead to specific precipitation fields. Our analysis indeed assumes that the precipitation intensity is not strongly modified and that no new emerging circulation patterns are formed yet. Concerning the former, our time series suggests that there is no substantial trend in the intensity at short time scale (1 hour, 1day). Concerning the latter, we did not see any report of the emergence of new patterns, but well of the possible change of frequency which is naturally taken into account by the selection of analogs, i.e. only analogs of the specific circulation pattern are in principle selected.

As you will see below in the text we underline the absence of substantial trends in extreme precipitation due to climate change, and we discuss the question of the modification of circulation patterns that could affect the frequency of events. Both points are now discussed in the text as follows in Section 5:

*"We acknowledge that the training dataset, which covers only a six-year period, is relatively short. It is a sophisticated merging of the radar datasets and the rain gauges data \citep{journee2023}. Moreover, a coherent coverage with the same radars started only in 2017. Efforts are currently ongoing at the Royal Meteorological Institute to extend this dataset for the following years and also to extract the best information from previous years.*

*The risk of misrepresenting climate change is an important problem that could affect the results. The question is twofold: On one hand the change of precipitation intensity and on the other hand, the change of circulation patterns. For the precipitation intensity, climate change has currently a very limited impact over Belgium as reflected by the long time series covering Belgium \citep{RMI1}. In the Uccle time series of 1-hour or 24-hour extremes, there is no*

*obvious trend (as for temperature for instance), except some long term modulation probably associated with low-frequency variability that still needs to be uncovered. In the recent attribution study of the extreme July 2021 case which affected the East of Belgium it was shown that there is no local trends of one-day or two-day precipitation accumulations, except the extreme July event, which appears like a black swan (Tradowsky et al, 2023). We therefore suspect that climate change does not affect much (currently) the extreme intensities. Therefore if there are changes in precipitation it should mainly be related to changes in circulation as illustrated on the website of the Royal Meteorological Institute (\cite{RMI2}, \cite{RMI3}). The analog approach should partly take that into account as the reconstruction of precipitation is done using analogs of weather situations (regimes) that were experienced in the past. So if the circulation regimes (but not necessarily their frequency) were the same in the training dataset and in the past, the method should appropriately take the change of frequency into account. As far as we can experience, there is no major disruption on the weather patterns (large scale pressure fields) yet that would imply that the analog approach cannot provide useful information on past weather situations. This question of change of circulation regimes is however important to address in the future due to the rapid changes experienced by our climate. Such an analysis will then be used to support further developments of the analog dataset."*

- Why was 2006 chosen for verification, and how do other years compare? Are the hourly precipitation fields representative or only the daily sums (as used in the verification)?

   (1) To substantiate why we chose 2006, the following was added to Section 3.2:

*"The year 2006 was selected as the case study year because from that moment onward, a considerable number of automatic weather stations became available, and the year was relatively wet. While more 'average' years such as 2007, 2012 or 2014 could have been considered, our focus on evaluating precipitation fields led us to prefer a relatively wet year."*

   (2) The verification focused on daily precipitation totals, as the analog matching was performed based on 24-hour accumulations. There was no direct comparison of hourly precipitation fields between the radar and historical days. Instead, the hourly distribution from the radar day was transferred to the matched historical day. Performing a verification at the hourly level would be computationally far more intensive and requires more processing time and memory.

- Please describe the methodology (in section 3) more precisely. Regarding the predictors: Why were these time steps selected? Why were these predictors selected? TEMP12 (one predictor), PRES012 (two predictors), RELHUM (three predictors at the same heights), TWS3 (three predictors at different heights), and TWS4 (four predictors) were compared with each other, and it was concluded that geopotential height provides the best results. It seems that the selection is very centered on the geopotential height right from the start.

Thank you for your feedback. To clarify this aspect, we changed the text in the beginning of Section 3 as follows:

*"The analog technique used in this study is based on the principle that local weather conditions are influenced by large-scale atmospheric patterns. According to the weather analog hypothesis, if two synoptic-scale situations are similar, they are likely to produce similar local weather outcomes. By comparing a given target day with past days that had similar atmospheric conditions, we aim to estimate local variables, such as precipitation. This approach assumes that similar large-scale atmospheric conditions lead to similar local precipitation patterns. While this assumption may not always hold perfectly, it provides a solid foundation for developing a high-resolution gridded precipitation dataset.*

*To identify analog days, we tested various combinations of atmospheric variables and time steps. The selection of predictors and time steps was guided by insights from previous research on analog methods. These previous studies have consistently shown that geopotential height is among the most effective predictors for identifying analog days, which is why it was prioritized from the outset \citep{horton2017, horton2019}. The results confirmed that sets including geopotential height consistently outperformed those without it.*

*Due to the computational burden of selecting analogs and validation when testing multiple variables across many pressure levels and time steps, only a limited selection of predictor sets was addressed in line with the previous literature. Therefore, we made informed, targeted selections to ensure feasibility, guided by a combination of trial-and-error and the findings from previous studies. Several combinations of fields and times of day were tested, and we ultimately arrived at a scheme that provides good results. However, we acknowledge that other combinations might perform even better, and further work is needed to optimize the method. These improvements are planned for the next version of the dataset."*

- It is not clear to me how many analogs (10 or 25) were used to compute the median for the published data set.

Thank you very much for your remark. 25 analogs were used to compute the median. It was mentioned in Section 4. ("The first archive is a database and contains a matching set of 25 best analog days."). To clarify this, we changed the text in Section 4 as follows:

*"The dataset is relying on the Zarr archiving format (Miles et al., 2020) and is composed of two such archives. The first archive is a database and contains a matching set of 25 best analog days for each target day from 1940 to 2016. The second archive contains the median hourly precipitation field of the 25 member ensemble (set of 25 best analogs for one day in the past) for every day from 1940 to 2016, calculated from the climatological RADCLIM radar product."*

- The data set can easily be read in with xarray, and I appreciate the sample code. However, the dataset, which must be downloaded and unpacked, is quite large at approx. 40 GB in zip format. Is there a reason zarr files with shorter time spans (e.g., one or ten years) were not created?

The data is stored in a single *zarr* archive to ease the download from Zenodo. Indeed, multiple Zenodo entries for the dataset would make it complicated to manage and curate, so we aimed for a single one, asserting that 40Gb is actually not a lot of data these days.

- An example precipitation field from the published data set would be good to include in the paper to give the reader an insight of the quality/variability of the precipitation field.

Thanks for the suggestion. A paragraph including a figure from both the observed precipitation field for a particular day as well as the median precipitation field – computed based on the 25 best analogs – was added to Section 4. So more concrete, the following is added:

*"Figure \ref{medianfield} presents an example precipitation field from the published dataset for 27.01.2001. The left panel shows the observed daily precipitation accumulation for that date (grid based), while the right panel displays the daily accumulation (sum) of the median of the 25 best analogs for that day. The published dataset includes such median-based precipitation fields for each day between 1940 and 2016."*

[Figure]

*Figure 15: Comparison of observed daily precipitation accumulation and the median accumulation derived from the 25 best analog days.*

**Specific comments**

- Section 1 Discussions and references to existing rainfall climatologies, such as EURADCLIM (https://doi.org/10.5194/essd-15-1441-2023) and RADKLIM (10.5676/DWD/RADKLIM_RW_V2017.002), are missing and would provide additional context for this data set.

A reference to both climatologies are added in the Abstract:

*"Similar products such as EURADCLIM (Europe) (Overeem et al., 2023) and RADKLIM (Germany) (Winterrath, 2018), both radar-based gauge-adjusted datasets, have already been created and published. Both datasets are high spatial resolution dataset (2 km and 1 km, respectively)."*

- Section 2.1.2 Please provide more information about the radar data set. The link to the Radclim user guide is insufficient. In a few years, this link may no longer be accessible, leaving the paper lacking further information on the radar data.

Thank you for raising this point. The text about RADCLIM has now been expanded as follows in Section 2.1.2:

*"The RADCLIM product is an offline counterpart to the real-time RADQPE (RADar Quantitative Precipitation Estimation) system \citep{ GenerationandVerificationofRainfallEstimatesfrom10YrVolumetricWeatherRadarMeasureme nts}. It reconstructs historical precipitation datasets using the same core methodology as RADQPE, but adapts it for retrospective analysis by incorporating missing radar or rain gauge measurements that were unavailable during real-time operations. This approach addresses the gaps and uncertainties inherent to instantaneous processing, resulting in a more complete and reliable dataset suited for climatological analyses.*

*RADCLIM provides high resolution radar-based quantitative precipitation estimation for Belgium and its surroundings. The RADCLIM product is obtained after a careful processing of the weather radar measurements and a merging with rain gauge measurements. This combination provides a detailed and accurate picture of precipitation distribution in time and space.*

*Data is collected through a network of weather radars and rain gauges. RADCLIM incorporates 3D reflectivity data from a network of four C-band dual-polarization weather radars: Jabbeke and Wideumont, owned and operated by RMI; Helchteren, managed by the Flanders Environment Agency (VMM); and Avesnois, operated by Meteo-France. The rain gauge data come from automatic networks of the Flemish Environment Agency (VMM), the Service Public de Wallonie (SPW), the Hydraulic Laboratory (WL), the Royal Meteorological Institute (RMI) and from manual rain gauges network CLIM of RMI. Both automatic and manual quality checks are carried out to ensure data reliability.*

*Radar reflectivity measurements are converted into rain rates. To get from raw radar reflections to reliable precipitation estimates, several processing steps are carried out. To mitigate non-meteorological echoes (e.g., ground clutter, wind farms), satellite cloudiness comparisons, vertical reflectivity profile analyses, and spatial texture detection are employed. The data from the different radars are then merged into a precipitation composite. Precipitation accumulations are calculated over 5-minute and 1-hour intervals using optical flow techniques. These data are combined with rain gauge observations every 5 minutes via a statistical interpolation method based on the hypothesis of Gaussian process. This creates a correction factor, which is then also applied to the 5-minute accumulations.*

*The final composite product has a spatial resolution of 1 km and uses the Belgian Lambert 2008 projection. Coverage extends from 0.3°W to 9.7°E and from 47.4°N to 53.7°N. Precipitation accumulations for 5 minutes and 1 hour are available based on UTC time and are provided in standard formats such as GeoTIFF and HDF5."*

- Section 2.1.3 Although the title indicates that this is daily rain gauge data, these data are not described in section 2.1.3.

This was indeed not very clear. The text in Section 2.1.3 has changed as follows:

*"Rain gauge data from the RMI's Automatic Weather Stations (AWS) have been available since 2003/2004. Prior to that, only 8h–8h precipitation data were recorded, which are not suitable when matching days based on 0h–0h accumulations.*

*RMI's weather stations are part of a global network that essentially records ground-based observations of temperature, humidity, air pressure, wind and precipitation, among other things. The RMI's weather stations are fully automated and are providing measurements every 10 minutes. Precipitation is measured by weighing rain. Based on the data from an AWS station, hourly accumulations can be provided."*

- Are only 2006 and 2017 to 2022 really available for verification?

Starting end 2003/begin 2004, data every 10 minutes and so, 0h–0h daily precipitation data became available for a limited set of stations (11 in total). As previously mentioned, other years after 2006 (i.e. 2007–2016) could have been used for verification, but 2006 was chosen based on the specific considerations outlined in General comment 1. The verification is extended to two additional years in the new Supplement.

- Section 3.1 Do I understand correctly that during the validation of the data set, a point measurement was compared with the average of the four nearest grid points of the radar data set? Please describe more clearly.

Yes, that's correct. Thanks for pointing out the ambiguity. This is now described more clearly in Section 3.1 as follows:

*"On the target day (in this case, a day in 2006), 24-hour precipitation measurements were available from 11 AWS stations. Once an analog day was identified using a given predictor set, the radar-based precipitation field for that analog day was used for comparison. Because radar grid points do not align exactly with the AWS station coordinates, we identified the four nearest radar grid points to each station. The mean precipitation from these four surrounding grid points was then used as the estimated 24-hour precipitation at that AWS location for the analog day."*

- Section 3.2 Why was the evaluation made for the location Uccle? What are the results for other locations?

The following text was added to Section 3.2 to clarify why Uccle was used for evaluation:

*"Just as Uccle serves as the national reference station in Belgium for monthly, seasonal, and yearly climatological reports, as well as for trend analysis, it was also chosen as the reference and evaluation location in this study."*

While other locations could certainly be included for precipitation which exhibits more local variability, geopotential height is a large-scale, smooth variable, varying only gradually over large distances, making local station choice less critical for that parameter.

- Section 4: The description of the actual data set is in general very short, the structure of the published data set could be described a little more. For example, I was not aware at the beginning that the first zarr data set only contains timestamps of the 25 analog days.

The text in Section 4 is now modified a bit. We also clarified to the reader with a footnote:

*"The dataset is relying on the \emph{Zarr} archiving format~\citep{zarr} and is composed of two such archives. The first archive is a database and contains a matching set of 25 best analog days\footnote{The database only contains the dates information, not the fields.} for each target day from 1940 to 2016. The second archive contains the median hourly precipitation field of the 25-member ensemble for every day from 1940 to 2016, calculated from the climatological RADCLIM radar product. The Zarr archives have been constructed using the \emph{xarray} Python library~\citep{xarray}, and are licensed under the Creative Commons Attribution 4.0 International license (CC BY 4.0).*

*The dataset is available on the Zenodo platform (\url{https://doi.org/10.5281/zenodo.14965710})~\citep{zarr_zenodo}. A notebook showing how to load and explore the dataset in Python is available at \url{https://github.com/ElkeDebrie/RADCLIM-Analogs}.*

*Figure \ref{medianfield} presents an example precipitation field from the published dataset for 27.01.2001. The left panel shows the observed daily precipitation accumulation for that date (grid based), while the right panel displays the daily accumulation (sum) of the median of the 25 best analogs for that day. The published dataset includes such median-based precipitation fields for each day between 1940 and 2016."*

[Figure]

*Figure 15: Comparison of observed daily precipitation accumulation and the median accumulation derived from the 25 best analog days.*

- L4-5 "wide-ranging" applications is very general, please give specific examples here.

Thanks for the note. Some examples are added in the Abstract:

*"An hourly high-resolution gridded precipitation product over Belgium can provide valuable insights into the dynamics of both short-term and long-term rainfall events, which can be used for wide-ranging applications such as flash flood forecasting and warning systems, studying precipitation extremes and trends, validating weather and climate models or detecting changes in precipitation patterns due to climate change."*

- L90-92 This sentence is part of the methodology and not data and may be confusing at first reading.

Indeed. Thank you for pointing that out. The sentence is removed and replaced by a description of the AWS rain gauge data (Section 2.1.3 and Specific comment 3):

*"Rain gauge data from the RMI's Automatic Weather Stations (AWS) have been available since 2003/2004. Prior to that, only 8h–8h precipitation data were recorded, which are not suitable when matching days based on 0h–0h accumulations.*

*RMI's weather stations are part of a global network that essentially records ground-based observations of temperature, humidity, air pressure, wind and precipitation, among other things. The RMI's weather stations are fully automated and are providing measurements every 10 minutes. Precipitation is measured by weighing rain. Based on the data from an AWS station, hourly accumulations can be provided."*

- L199-200 Why were these threshold values chosen?

The threshold values were chosen because they are location-specific percentiles derived from the empirical distribution of 24-hour precipitation accumulation at the selected reference location, Uccle. This approach ensures that the thresholds reflect the local climatology and capture a representative range of precipitation intensities—from dry days to heavy rainfall events. By selecting percentiles rather than fixed values, the ROC analysis remains adapted to the precipitation characteristics of the location, allowing for a more meaningful evaluation of performance across the full spectrum of rainfall amounts. This method avoids bias that could arise from applying arbitrary or non-representative thresholds. This description is provided in Section 3.2.3:

*"The selected threshold values are location-specific and are determined based on percentiles derived from the actual distribution of 24-hour precipitation accumulation for the specific location, here Uccle. From top right to bottom left, the threshold values used in the ROC curves in Figure 8 are: 0 mm, 0.1 mm, 0.2 mm, 0.54 mm, 0.8 mm, 1.78 mm, 3.6 mm, 4.74 mm, 6.66 mm, 9.98 mm, 27.3 mm."*

- L213 How many precipitation events exceeded 15 mm/24 hours in 2016?

I suppose you mean 2006? There were 4 events in 2006 exceeding 15mm/24h. It is indeed good to mention the number of events, as there were not many.

This is added in Section 3.2.3:

*"Figure \ref{diff_rocs1} presents probabilistic forecasts generated from the 10-member analog ensemble for 2006 at Uccle -- only 4 events in 2006 exceeded 15mm/24h, which is not much and why it is useful to consider a longer period of time."*

- L245ff Figure 14 and the two locations without monitoring points do not provide much additional information. I suggest removing Figure 14 and this sentence.

We agree with the remark. We chose to keep the sentence, but to move the Figures to the supplementary material.

- Figure 15b Is the hourly precipitation data from a radar or a rain gauge? Section 2 only introduced daily rain gauge data.

(Related to Specific comment 3 & 8) "Daily" is removed in the title and a better description of AWS stations is added in Section 2.1.3.

*"Rain gauge data from the RMI's Automatic Weather Stations (AWS) have been available since 2003/2004. Prior to that, only 8h–8h precipitation data were recorded, which are not suitable when matching days based on 0h–0h accumulations.*

*RMI's weather stations are part of a global network that essentially records ground-based observations of temperature, humidity, air pressure, wind and precipitation, among other things. The RMI's weather stations are fully automated and are providing measurements every 10 minutes. Precipitation is measured by weighing rain. Based on the data from an AWS station, hourly accumulations can be provided."*

- L269-270 Please describe/justify this statement.

This explanation is added to the statement in Section 4.

*"If many analog days have zero precipitation, using the median of their precipitation values can be misleading—it will tend to be zero or very low. This means higher precipitation amounts won't be well represented, even if a few analog days had significant rainfall. In such cases, using the mean instead of the median can help. The mean takes into account all values, including the higher ones, so it can better reflect the presence of heavier precipitation within the set of analog days—even if most of them were dry."*

**Technical comments**

- Figure 1 There is no reference to this figure within the text. A reference to the figure is made in the Introduction.
- Figure 2 Please add the x and y axes to the left figure. The figure under consideration is replaced.
- Figure 3 Please add units to the y-axis. Done.
- Figure 15 Please add the unit to the x-axis. Thank you. Lag in hour was added to the x-axis.
- Line 201 The reference Wil (2019) is missing in the reference list. Thank you for pointing that out. Done.